# Workload, Job Satisfaction and Quality of Nursing Care in Italy: A Systematic Review of Native Language Articles

**DOI:** 10.3390/healthcare11182573

**Published:** 2023-09-18

**Authors:** Nicola Serra, Stefano Botti, Assunta Guillari, Silvio Simeone, Roberto Latina, Laura Iacorossi, Martina Torreggiani, Monica Guberti, Giancarlo Cicolini, Roberto Lupo, Angela Capuano, Gianluca Pucciarelli, Gianpaolo Gargiulo, Marco Tomietto, Teresa Rea

**Affiliations:** 1Biostatistics Unit, Department of Public Health, University Federico II of Naples, 80138 Naples, Italy; nicola.serra@unina.it; 2Hematology Unit, Azienda USL-IRCCS of Reggio Emilia, 42123 Reggio Emilia, Italy; 3Department of Public Health, University Federico II of Naples, 80138 Naples, Italy; assunta.guillari@unina.it (A.G.); teresa.rea@unina.it (T.R.); 4Clinical and Experimental Medicine Department, “Magna Graecia” University, 88100 Catanzaro, Italy; silvio.simeone@unicz.it; 5Department of Health Promotion, Mother and Child Care, Internal Medicine and Medical Specialities, Università degli Studi di Palermo, 90133 Palermo, Italy; roberto.latina@unipa.it; 6Nursing Research Unit IFO, IRCCS Regina Elena National Cancer Institute, 00144 Rome, Italy; laura.iacorossi@gmail.com; 7Research and EBP Unit, Health Professions Department, Azienda USL-IRCCS of Reggio Emilia, 42421 Reggio Emilia, Italy; martina.torreggiani@ausl.re.it (M.T.); monica.guberti@ausl.re.it (M.G.); 8Department of Precision and Regenerative Medicine and Ionian Area—(DiMePRe-J), University of Bari “Aldo Moro”, 70124 Bari, Italy; giancarlo.cicolini@uniba.it; 9San Giuseppe da Copertino Hospital, Local Health Authority, 73043 Copertino, Italy; roberto.lupo@uniba.it; 10Department of Emergency, AORN Santobono-Pausilipon, 80122 Naples, Italy; angelacapuano2016@gmail.com; 11Department of Biomedicine and Prevention, University of Rome Tor Vergata, 00133 Rome, Italy; gianluca.pucciarelli@uniroma2.it; 12Hematology and Haematopoietic Stem Cell Transplantation Centre, University Federico II of Naples, 80138 Naples, Italy; gianpaolo.gargiulo@unina.it; 13Department of Nursing, Midwifery and Health, Faculty of Health and Life Sciences, Northumbria University, Newcastle upon Tyne NE1 8ST, UK; marco.tomietto@northumbria.ac.uk

**Keywords:** nursing workload, job satisfaction, quality of care, native language, Italian

## Abstract

Nursing research is rapidly increasing, yet contributions from numerous countries that may interest the international nursing community are impeded because many research articles are published in authors’ native language and not in English. The objectives of this work were to systematically review papers published in Italian related to job satisfaction and the quality of nursing care, and to discuss their findings in light of the international literature. The Preferred Reporting Items for Systematic Reviews and Meta-Analyses (PRISMA) method was used. The Directory of Open Access Journals (DOAJ) and Indice della Letteretura Italiana di Scienze Infermieristiche (ILISI) databases were consulted for eligible studies published from January 2015 to November 2022. Two hundred sixteen papers were identified, 11 of which were selected for review: 8 on job satisfaction, two on workload issues, and 1 on quality of nursing care. The quality of included studies was assessed through the Effective Public Health Practice Project quality assessment tool (EPHPP). The results of our review were in line with those of international literature, and they can help to fill the knowledge gap on the quality of nursing performance in Italian care settings. In addition, the proposed method can provide further elements of discussion among literature providers and reviewers.

## 1. Introduction

Nurses constitute the largest category of healthcare professionals in almost all EU countries. Their crucial role in providing care in hospitals, long-term care facilities and the community has been highlighted again during the COVID-19 pandemic. The demand for nurses will continue to increase in the coming years due to an aging population, as many nurses are approaching retirement age. Increasing the retention rate of nurses is a growing concern to avoid exacerbating current and future shortages. Concerns about growing shortages have prompted many countries to increase the number of students in nursing education programmes, although it will take some years for the impact to be felt. On average, there were 8.3 nurses per 1000 population across EU countries in 2020 [1].

Regarding the increase in the chronically ill population compared to the data of the National Federation of Italian Nurses (FNOPI), 63,000 nurses are missing in Northern and Southern Italy. Although nurses are more numerous than average, according to international standards, there are only three nurses per doctor. The personnel shortages are known to the Italian Government, yet despite having increased the number of places in university training, to date, an adequate number of nursing personnel has not been obtained [2].

There are even disparities in terms of pay. The salary of nurses refers to the average gross annual income, including social security contributions and income taxes. Most countries’ data are based on registered nurses working in hospitals (“professional”). The median income of nurses is below the European average of all nurses and is on par with Portugal and Greece [1].

Furthermore, the European nurse ratio is 8.3/1000 inhabitants, and in Italy today, it is 6.3/1000 inhabitants; there needs to be a clear distinction between the different types of nurses and hospitals, and no National reference standard is available. The model for determining this ratio considers some parameters related to the complexity of care, the clinical stability of the patients and the nursing commitment of non-self-sufficient patients [2].

The importance of nursing has emerged over the last decade thanks to the spread of research findings that nursing has been fundamental for clinical practice development.

Along with the expanded professional role of nurses in healthcare settings, the availability of nursing evidence has significantly increased [3,4]. Several papers have been published in national journals in the authors’ native language only. However, this may reduce the international visibility of research results [5]. For various reasons, including but not limited to language barriers, many Italian researchers and/or research groups prefer to publish their studies in their local language, as observed in other countries [6,7]. Italian nurses have been showing strong interest in publishing results, information, analyses, and opinion papers in their language to meet the increasing demand for evidence by national readership and policymakers [8]. The intention of showing their findings in their language could be associated with the local relevance of the findings, the type of reader targeted, or unsuccessful attempts to publish in other journals, including international journals [6,7].

The role of the target culture was highlighted by some authors [9,10] in the context of analyzing the effects of cultural issues on readers [11]. Additional challenges included technical difficulties, vocabulary limitations, the stress experienced by the author caused by the review process in a non-native language and cultural nuances of reviewers [12].

In this paper, we investigated some aspects of native and international literature, focusing on the quality of nursing care and job satisfaction.

The quality of nursing care has been linked to three main factors: workload, quality of perceived nursing care, and job satisfaction [13].

Nurses’ “workload” was variously defined depending on what concepts were considered within its definition and the stakeholders (patient, nurse, manager) interested in its consequences [14]. Authors defined it as the amount of work a nurse performs within a specific period, the number of tasks required, the work carried out on patients and administrative tasks [15,16,17]. Qualitative aspects such as the demand for nursing care, nursing skills, patient complexity, intensity of care, and patient self-care level were added by other authors [18,19,20,21,22].

In a concept analysis, nursing “workload” was defined into five main categories: the amount of nursing time, proportion of direct patient care, the level of nursing skill, the amount of physical exertion, and the complexity of care [14]. Physical and psychological issues also condition the nurse’s performance quality during work. Physical workload depends on the nurse’s activities understood as the amount of physical work performed. In contrast, the psychological workload is determined by nurse’s mental efforts such as managing information, relationships, and making decisions. As reported in the literature, all these elements are strongly correlated and should be considered in measuring nurses’ workload [23].

The quality of nursing care is linked to the patients’ satisfaction with the care provided by the nurses and the information received [24]. Literature has shown that patient’s satisfaction was linked to improved treatment adherence and better outcomes [25]. Satisfied patients were more likely to recommend to other people the health institution where they were cared for; health institutions particularly welcomed this to improve their attractiveness [26].

For the above reasons, patients’ satisfaction has been considered the best quality of care-related index collected by health institutions [27], and quality of nursing care perceived by the patient is a fundamental component of quality management, healthcare planning and evaluation systems [26,28].

Nurses’ job satisfaction is a significant component affecting patient’s satisfaction, and it is influenced by multiple factors, including adequate duration of daily shifts and their frequency, remuneration, amount and complexity of work, own attitudes of workers [29,30]. Furthermore, it was demonstrated that a supportive workplace culture positively influenced professionals’ satisfaction and outlook, while negatively impacting the intention to leave the job. Conversely, work–life imbalance decreased satisfaction and outlook, while increasing turnover intention [31]. Job satisfaction is crucial to prevent burn-out. It is defined as a “syndrome resulting from chronic workplace stress that was not properly addressed” [32], leading to labor shortages, a high rate of leaving work and turnover, a negative effect on patient care, and associated costs [33]. Nursing management intervention should be implemented in healthcare institutions to enhance nurses’ practice, and improve nursing quality and patient satisfaction [13]. This work aimed to review available papers on Nursing Workload (NW), Job Satisfaction (JS) and Quality of Care (QoC) published in the authors’ native language (Italian) and to discuss the findings in light of the international literature. This could provide visibility to studies whose results on the quality of nurses’ performance may have greater dissemination and recognition.

## 2. Materials and Methods

We conducted a systematic review following the Preferred Reporting Items for Systematic Reviews and Meta-Analyses, the PRISMA statement [34].

The study protocol has been stored at the following link: “https://archive.org/details/osf-registrations-hn3bf-v1 (accessed on 5 August 2023)”, with registration doi https://doi.org/10.17605/OSF.IO/HN3BF. The research question for this systematic review was: “How many Italian articles have been published on job satisfaction, workload and quality of nursing?”.

### 2.1. Inclusion Criteria and Eligibility

The literature review process was conducted in the following steps: identification of the research questions through the PIOS (Population, Intervention, Outcome, Design) method, literature search, included papers selection, findings appraisal and summary building.

Eligible studies should meet the following criteria: (1) population: nurses who work in both hospital and community settings; (2) intervention: the measurement of different levels of outcome; (3) outcome: quality of nursing care, workload, job satisfaction; and (4) design: primary studies whose reports aimed to assess perceived quality of nursing care, workload, and job satisfaction. Theses, posters, commentaries, letters to the editor, review papers, and books were excluded as well as papers whose full text was not available.

### 2.2. Search Strategy

The search was performed on the Directory of Open Access Journals (DOAJ—https://doaj.org/, accessed on 5 August 2023) and the Indice della Letteratura Italiana di Scienze Infermieristiche (ILISI—https://ilisi.opi.roma.it/, accessed on 5 August 2023) databases, with language restriction (Italian), using and updating (from January 2015 to November 2022) the search strategy reported in Table 1. The DOAJ database indexes all scientific journals with a quality control system to guarantee the published content and records papers in all native languages. ILISI is a literature repository powered by the Italian health profession order (FNOPI) with the aim of contributing to the enhancement and diffusion of nursing literature published by Italian journals. The quality control system of both databases includes an International Standard Serial Number (ISSN), an editorial board, publication frequency information, a peer review process, a content control system, information on journal publication policy and ethics, and information for users.

### 2.3. Study Selection and Data Extraction

Italian search terms were adopted. To improve the validity of the search strategy, the identification of keywords, building search strings, and the electronic database navigation were performed by three experts together (a biostatistician and two research nurses with more than 10 years of experience). Two independent authors (N.S., S.B.) screened titles and abstracts according to the search strategy. Following the first phase, they independently assessed the full text of all potentially relevant studies for inclusion in this review. Any disagreement was resolved through discussion with a third author (S.S.). Then, using a standardized data collection form, the following information was extracted from the included studies: 1st author, journal, publication year, title, database, study aim, type of study and sample size, and results.

### 2.4. Quality of Evidence

The quality assessment of included papers has been performed independently by two reviewers, a nursing researcher with expertise in reviewing methods and a biostatistician with expertise in critical appraisal, using the Effective Public Health Practice Project Quality Assessment Tool (EPHPP) [35]. This tool was chosen because it was developed for systematic reviews of public health topics and can be used across multiple study designs. The tool considers six domains for the assessment: “selection bias”, “study design”, “confounders”, “blinding”, “data collection methods”, and “withdrawals and drop-outs”. Weak, moderate, or strong quality level was attributable to each dimension and global score. A standardized guide and dictionary are available for the rating system (Table 2).

The assessment result was compared and discussed between the reviewers until a final decision on the global score was reached. Unresolved conflicts would be managed involving a third reviewer.

### 2.5. Data Synthesis

The feasibility of performing statistical analyses and meta-analyses for each review topic was evaluated. Alternatively, the review findings were discussed among the research group to provide a narrative report; then, the review findings were compared and discussed in light of those provided by the international literature.

## 3. Results

Two hundred and sixteen records published between January 2015 and November 2022 were found, 65 from ILISI and 151 from the DOAJ database. Two duplicates were removed before screening.

Of the 214 records screened, 199 were excluded after titles and abstract reading for the following reasons: 140 did not meet the inclusion criteria, 55 were published before 2015 and four were written in English. Fifteen reports were reviewed for eligibility, four of which were excluded due to being reviews (three reports) or a book (one report). Eleven reports were included in our review (Figure 1), six from the ILISI database and five from the DOAJ database.

Eight papers had JS as a primary outcome, two NW and one QoC. Nine were cross-sectional studies (six multicentre and three monocentre), one used a mixed method approach, and one had a longitudinal design. A meta-analysis was not feasible due to the heterogeneity of study designs, recruited populations, outcomes, measures, and settings. A narrative description of the included studies was provided, and the findings were summarized in Table 3. The quality assessment of included studies confirmed what other authors found in papers published in Italian journals [36]. The study methods and statistical analyses were simple and the selection bias was present in all included papers. In many studies, data collection has been performed using unreliable methods such as an unvalidated “ad hoc” questionnaire frequently shared online through free web platforms. The level of evidence provided by these studies was weak. The results of papers’ quality assessment are summarized in Table 4.

### 3.1. Nursing Workload

A perspective observational study [37] assessed NW in a pediatric Intensive Care Unit (ICU) collecting the daily Nursing Activities Score (NAS) [48] for two months. Four hundred and fifteen observations were collected on 51 patients admitted during the study period. According to the NAS, 4% of patients resulted having very high complexity for caring, 25% high complexity, 51% median complexity, and 20% low complexity. The monthly mean score varied from 65.35(SD ± 8.19) to 66.25(SD ± 6.34). NAS per diagnosis at hospital admission ranged from 47.3 (vascular shunt) to 152.7 (anaphylactic shock) and the ratio of nurses per patient ranged from 1:1 to 1:1.4.

The second paper [38] aimed to survey nurses’ workload during the COVID-19 pandemic. Perceived care complexity was measured combining the Dependency to Care Index (IDA) [49] and some parts of the Index of Caring Complexity (ICC) [50]. Based on the adopted score, 36.3% of nurses perceived a high level of workload, 49.1% perceived a median level and 14.6% a low level. Nursing activities that had mainly influenced NW were patients’ hygiene and ensuring comfort (mean IDA score = 1.6), mobilization (mean IDA score = 1.7), and safety (mean IDA core = 1.9).

### 3.2. Quality of Care

Only one paper found had the QoC as the primary outcome [39]. The family satisfaction with advanced cancer care-2 tool (FAMCARE-2) [51] was used to assess QoC perceived by 150 caregivers (CGs) of patients who died in a palliative care setting. The number of CGs satisfied with safeguarding the dignity of patients were 147 (98%), those satisfied with the healthcare providers’ availability were 144 (96%), 132 (88%) were satisfied with the quality of information received, and 132 (88%) with daily primary care provided for their loved ones.

### 3.3. Job Satisfaction

All studies included in this section were cross-sectional. In a study [40] on a cohort of 115 nurses working in various hospital settings, 88 (76%) were unsatisfied with their job conditions regarding organizational well-being and pay. Eight of 10 scores on a 1–10 Likert scale were assigned by 97 (84.4%) participants assessing the complexity of their work as well as the same score was assigned to their interest in nursing work. High variability was reported on nurses’ autonomy; 75 (65.3%) of the respondents reported moderate to very low autonomy on prioritization of their activities, while 37 (32.1%) reported high to very high levels of autonomy. Less than half of the sample (56; 48.7%) reported very low to moderate autonomy in their work. Depression was reported by 40 nurses (34.8%). Anxiety, fatigue, and irritability were reported by 86 (74.8%), 103 (89.6%), and 99 (86.1%) nurses, respectively. Cosentino et al. [41] evaluated the JS through the McCloskey Mueller Satisfaction Scale (MMSS) [52] in a cohort of nurses working in emergency areas or high care settings. JS resulted in generally low scores; only two JS-related domains were a little over the 3.5 mean score, which means “not satisfied and not unsatisfied” on a 1–6 Likert scale (interaction with colleagues mean score = 4.0; social interaction mean score = 3.6), while other JS-related domains ranged from mean scores of 2.5 to 3.3. The multiple regression analysis identified a model including four main predictive factors of JS; colleagues’ collaboration impacted on JS at 20%, head nurse leadership style at 10%, compassion satisfaction at 3.4% and autonomy at 0.9%. In another study [42], the JS of nurses working in various inpatient words (internal medicine, surgery, emergency) was assessed using the Stamp’s Index of Work Satisfaction (IWS) [53]. Two factors significantly influenced the IWS score: the intention to leave work for 3 years (*p* < 0.01) and type of work contract (open-ended or fixed-term) (*p* < 0.01). Of 264 participants, 56 (21.2%) reported the intention to leave their work within 3 years, 141 (53.4%) had no intention and 67 (25.4%) were unsure. Autonomy, professional status, task requirements and interaction with colleagues appeared to be the main components of work-leaving decisions. Most participants (233; 88.2%) had an open-ended work contract, while 31 (11.8%) had a fixed-term contract. The latter was a main dissatisfaction factor because it resulted in professional uncertainty and less pay. In a cross-sectional study [43], a psycho-social, Likert scale-based (1 to 5), “ad hoc” questionnaire was used to evaluate JS components, such as relationship with colleagues, leadership style, task characteristics, and pay, in a composite sample of physicians (37%) and nurses (63%) before and after the transfer of their entire hospital ward from one hospital to another. The authors concluded that no differences in JS were recognized, only evaluating mean scores obtained from each item. No statistical tests assessed the relationships between variables or their significance. Another paper explored factors influencing young nurses’ intention to leave medical wards and move to other care settings [44]. An “ad hoc” questionnaire was administered to 144 nurses under 30 years of age working in various hospitals in Italy. Fifty participants (35%) were delighted with their work, 86 (60%) were satisfied, eight (5%) were unsatisfied and none were very unsatisfied. Twenty-nine (20%) nurses manifested the intention to move to the emergency department (14; 48%), surgical ward (7; 23%) and other wards (8, 29%). The main factors influencing the decision to leave the medical ward were qualitatively recognized as the lack of professional recognition and growth, teamwork quality, pay, workload, and autonomy. Exploring JS, a mixed method approach was used by Incarbone and colleagues [45] who administered the Italian version of the Job Satisfaction Scale (JSS) [54] to 133 nurses working in three different settings (surgical, medical, and emergency areas). Eight semi structured interviews were carried out to improve participants’ experience understanding through a phenomenological approach. Eighty-one nurses (61%) were satisfied with their work, while 28 (21%) were unsatisfied and 24 (18%) were unsure. The qualitative analysis merged seven macro-areas: the higher intention to change of workers aged from 45 to 50 years versus greater resistance of those over 50 years; different grading of JS through the various settings; external and internal satisfaction correlated to intention to change or not, respectively; and the lack of intention to change associated with unsatisfying relationships and/or general dissatisfaction. Factors associated with poor JS were explored by a multicentre cross-sectional study that enrolled 120 nurses working in different settings of various hospitals [46]. In an “ad hoc” questionnaire, 95 (79.1%) respondents perceived excessive work responsibilities, 86 (71.6%) perceived a lack of autonomy, 102 (85.0%) reported doing something beyond their professional role and 96 (80.0%) reported a high workload. Low JS was perceived by 54 (45.0%) for poor recognition of their role, 51 (42.5%) for fatigue, 45 (37.5%) for pay and 14 (11.6%) for relationships with colleagues. A MMSS mean global score of 3.43 (±0.75) was reported in another multicentre cross-sectional study that investigated Head Nurses’ (HN) JS in three hospitals in Rome [47]. The eight dimensions of the MMSS ranged on average from 2.62 (±1.1) in the “extrinsic rewards” dimension and 3.66 (±0.81) in the “coworker” dimension. The details of the included papers are summarized in Table 3.

## 4. Discussion

About 80% of journals indexed in SCOPUS have been published in English [55]. While recognizing the importance of having a common language to exchange information and results of biomedical research from a worldwide perspective, its extensive use has created barriers for non-native speakers. For example, paper preparation for journal submission using English resulted in more difficulty for authors for whom English was a foreign language [56]. In addition, the lack of knowledge of the English language may lead to severe limitations on evidence accessibility [57]. Therefore, strategies to improve the efficacy and dissemination of scientific findings by overcoming linguistic (and cultural) barriers appeared mandatory to ameliorate health literacy [58]. This work will review papers available in the Italian language that may contribute to improving knowledge on interesting nursing topics such as NW, QoC, and JS. Discussing nursing research findings from these papers could be an essential opportunity to improve the diffusion of studies not accessible to international readers. Moreover, most of the systematic reviews do not include language-specific evidence. This generates a bias and a need for context-specific evidence to inform policymakers and the scientific community. The overall quality of included studies was low, and their high heterogeneity impeded any statistical analysis or comparison. The findings were discussed with the whole group and described here narratively. However, a discussion was provided considering the findings of international literature. This allowed us to highlight the need to improve research in these fields and disseminate (and/or translate) its results in the Italian language and better understand the processes affecting the quality of working life of Italian nurses. The topics considered are susceptible to many factors, including the national and local context in which they are studied. Providing a summary of the results of studies that would not be selected for international systematic reviews could be informative.

### 4.1. Nursing Workload

NW was strongly associated with the complexity of care and patient’s clinical conditions at admission [59] and during ICU stay [60]. Both physical and mental workload are generally high in ICU setting with a significant relationship between them. It was demonstrated that interventions aimed at reducing physical workload have had effects on mental workload and vice versa [23]. Increasing the NW impacted on quality of care, job satisfaction, and nurses’ burnout symptoms [61]. In addition, a variety of factors, including nursing staff proportion, resources allocation modality, and specific and/or unpredictable situations increasing health care needs (e.g.,: COVID-19 outbreak), influenced patient outcomes in various settings including both pediatric and adult ICU ones [23,29,62]. In light of these findings, the routine use of tools for the assessment of NW during nurses’ daily practice may provide useful information for nurses’ allocation in any circumstances, as reported by the two studies included in this review [37,38]. The relationship among NW and entry diagnosis in ICU settings was highlighted here; however, this finding was still largely under-recognized in the literature [23]. The assessment of care engagement should be used to adjust staffing (both numerically and as a skill mix). To be able to do this, however, the data must be reliable and used in real time. In a study, the three systems used provided noncomparable situations [63]. The authors concluded that only the implementation of outcome surveillance systems may allow collecting information to understand the effects of the number (and qualification) of personnel, and demonstrate the benefits of a workforce consistent with the workload or the damage deriving from a number of insufficient nurses compared to the care load. Surveillance of indexes and/or sentinel events (and patient satisfaction), using administrative data that are increasingly easily available at an Italian level, can be useful for choosing the minimum data sets, to guide the evaluation of whether and how much, at different complexities/care commitments, appropriate responses are ensured [63].

### 4.2. Quality of Care

The quality of nursing care is of paramount importance to the health and well-being of patients. Nurses play a crucial role in the healthcare system, as they provided direct care to patients and ensure effective monitoring and management of their healthcare needs [64,65]. The quality of nursing care can be evaluated based on several criteria: clinical competence, patient safety, treatment efficacy, patient empathy and care, respect for the patient’s dignity and rights [66]. The quality of care can be measured based on the satisfaction of patients and families [67]. Patient satisfaction with received care is considered the best quality outcome of care and an outcome of healthcare services [27]. Patient satisfaction measurement provided crucial performance information, thus contributing to total quality management. Concerning the quality of nursing care, our review included only one study conducted in a palliative setting [39]. Its findings aligned with those obtained by other studies in similar care settings [68,69,70]. They were consistent with recent guidelines, underlining that a personalized, multidisciplinary approach to the end of life care represented the most effective strategy to meet patient and family complex needs [71,72]. Evaluating and ensuring the quality of nursing care requires an ongoing effort by healthcare institutions to provide adequate resources, ongoing education, supervision, and support to the nursing workforce [73]. Furthermore, the adoption of evidence-based guidelines, performance monitoring and the active involvement of nursing staff in continuous improvement can help ensure a high standard of care and satisfaction for both patients and nurses [29].

### 4.3. Job Satisfaction

The main factors improving nurses’ JS were the perception of the high quality care provided and a climate of good teamwork. At the same time, those worsening it were high workload, unsupportive managers, staff shortages and low recognition and pay [74]. In the nursing field, the perception of organizational structural empowerment is assumed to be fundamental for a serene and continuous work performance; in contrast, the lack of motivation and power can be a negative factor predisposing nurses to burnout [33]. The climate is a factor that determines the performance of the organization and, as such, must be explored and optimized. Its importance is highlighted in the concept of work stress. Occupational stress derives not only from environmental factors, but also from the subjective assessment of the stressors (stressful factors) carried out by the individual worker. An assessment is also influenced by the socially shared subjective perceptions of those working in the same organization. It has been shown that work-related stress is influenced by professional status, which in turn is influenced by organizational climate, work organization, personnel management methods, and level of job satisfaction [75]. An element not to be underestimated and that contributes to improving the work climate, is the leadership style, which, if appropriate, favors greater job satisfaction among the nursing staff with positive effects on the quality of the assistance provided [76]. Utriainen and Kyngas [77] proposed a framework of factors influencing JS divided into three categories: (1) interpersonal relationships, (2) patient care, and (3) organizing nursing work. Several factors, such as relationships with coworkers, the feeling of togetherness, interaction and communication, teamwork, social climate and ethics, and peer support, conditioned relationships between hospital professionals. The significance of patient care to nurses, the quality of human relationships, and the perception of providing high quality care influenced the patient care dimension. Finally, various factors influenced work organization, such as leadership, the relationship among the team and the patient’s family, workload, work environment, nursing practice approach, pay, benefits, work autonomy, and professional development. The directions of the effects on JS of each single factor were known, while the magnitude of their combinations is largely unpredictable. De Simone and Siani [43] investigated the perceptions of hospital doctors and nurses regarding organizational policies, practices, procedures, and behaviors that most influenced work climate. The authors assessed JS variations before and after big organizational changes (e.g., ward transferred to another hospital). Relational characteristics such as the quality of the interaction between staff members and the leader–collaborator relationship resulted being the dimensions with the greatest influence on healthcare professionals’ JS. Organizational change did not lead to significant variations in work satisfaction due to the presence of engaging and participatory leaders as a result of this study. The preventive sharing of the project, its objectives and related processes by collaborators appeared to be a critical success factor. Trust in the leader was decisive during the sharing and agreement process [43]. Similarly, Cosentino and colleagues [41] concluded that the job satisfaction of nurses in critical care areas was mainly influenced by relational factors, which should be improved. The study highlighted some critical points for future research, particularly the association between job satisfaction and professional autonomy. Professional autonomy may have various meanings for study participants, such as the free application of their protocols, using their skills and competences for the decision-making process, or both. Knowing what the participants meant to understand better through collaborative dynamics would be interesting. Although many authors indicated the quality of inter-professional collaboration as a main factor influencing JS, the environmental perception of collaborative quality may vary among individuals and professional profiles [78]. Staff nurses involved in the care process were led by various figures (e.g., head nurse, senior nurse, nurse director) with different leadership styles, depending on their education and experiences. The nursing staff recognition by the leaders was a critical factor influencing various aspects of JS, including working time perception, relationships with colleagues, professional opportunities, social interaction, praise, and others [79]. However, head nurses’ own JS was influenced by structural empowerment, especially some dimensions such as social and professional interaction, praise and recognition, and control/responsibility [43]. As reported by another included study [47], head nurses’ JS was low, and their perceived structural empowerment was even lower, against the increasing volume of work, tasks, and performance expectations [80]. Burnout and nurses’ engagement in health services were also influenced by the quality of leadership [81]. Organizational well-being played a fundamental role in nurses’ work experience, contributing to improving performance [79] and quality of health services and patient outcomes as a consequence [82,83]. The nursing shortage is an unresolved, worldwide issue [84]. This indicates that something did not work in the last two decades in the development of the nursing role, and the high turnover of nurses among various care settings is a symptom of this situation. Outcomes such as nursing turnover and the intention to leave work were well documented by the international literature [85]. Twenty to forty percent of newly licensed nurses leave their work within the first year of activity [86,87]. Young nurses appeared more inclined to leave work, changing their care setting or quitting more frequently than older nurses [44,45,88]. Nurses were confident of working in a “helping environment” where they obtained satisfaction automatically by dealing with patients’ needs. However, various factors, including long shifts, tiredness, menial tasks, and relationship issues with physicians or patients, “brought young nurses back to the earth” and posed dissatisfied conditions leading to job choice questioning [89]. Although the decision to leave work depends on many factors, including personal, professional, and organizational ones [33], barriers and facilitators for implementing residency programs for nurses should be considered [86]. These programs were demonstrated to increase nursing residency (from 70% to 98%) and to reduce associated costs when implemented within the organization [90]. Considering that nurses’ JS and its consequences, such as turnover, burnout, and worst working climate, were associate with worst patient outcomes and increasing costs, nursing managers should implement pathways for the early recognition of factors influencing these phenomena to plan adequate interventions [33]. Strategies such as the optimization of internal communication [85], the improvement of newly graduated nurses’ organizational adjustment [91], and the improvement of career opportunities, extrinsic rewards, scheduling, interactions and support, praise and recognition, work environment, and hospital systems [92], should be considered. Our review reported differences in JS depending on the care setting, with nurses working in the emergency department appearing more satisfied than those working in the surgery [45]; however, this was not confirmed by the international literature. Perceived JS can vary during the nurses’ working life due to the continuous evolution of working conditions that could include changes in care settings, leaders, tasks, the level of matching between work and private life, and other factors [88]. Nurses with fixed-term contracts experienced work uncertainty and low pay. The salary was a fundamental recognition for the employees in the Italian context, as it was perceived as a consequence of their work rand represented the main tool for their livelihood [33,93]. In our review, nurses considered it inadequate relative to their expectations, needs and professional responsibilities [40,42,44]. Health institution managers should consider this carefully to limit undesirable behaviors, such as the reduction of work performance, absenteeism, and work quitting [94,95]. The theme of the distance between nursing management directions and nursing staff needs appeared as a transversal issue in our review. All included studies referred to it and concluded there was a need for tools and pathways to recognize, measure, assess, and monitor JS to improve organizational and clinical outcomes and costs. As reported by Ziello et al. [42], the Italian Ministry of Public Function promoted directives for the JS assessment in public organizations to promote strategies to improve employees’ quality of life and level of wellness. Nurse managers may influence a variety of factors that enhanced JS, such as but not limited to, ensuring support, decentralization, increasing specialization, informal relationships, allocating resources to acceptable levels, decreasing workloads, and involving nursing staff in policies that affect them [96].

### 4.4. Strenght and Limits

This systematic review brings new knowledge in summarizing the state of the art on workload, job satisfaction and quality of nursing care in Italy. Its results can inform nursing managers and nurse policy making on the multifactorial nature of organizational well-being and, therefore, on the need to guarantee the identification of health risk factors, a form of economic recognition, and an orientation path to support this health for nursing staff figure, which enhance its role in the various healthcare contexts for a better response to the needs of users. Moreover, this review has some limits. Firstly, using only two databases may have highlighted some of the Italian literature in the considered timeframe. Some of the nursing research journals surveyed in Italy may have been excluded. The results of this review should therefore be deepened with other subsequent works that consider this aspect to limit the studies’ selection bias.

## 5. Conclusions

The review revealed that the works published in Italian are few. Despite this, the study made it possible to highlight how the Italian nursing population is in a difficult situation with workload and job satisfaction. In light of the paucity of studies aimed at understanding the complexity of care, the need emerges to modify the care and organizational paradigms of nursing presence in various settings, both proportional to the type of care needs and to protect organizational well-being, with positive implications for the quality of care perceived by staff and patients.

## Figures and Tables

**Figure 1 healthcare-11-02573-f001:**
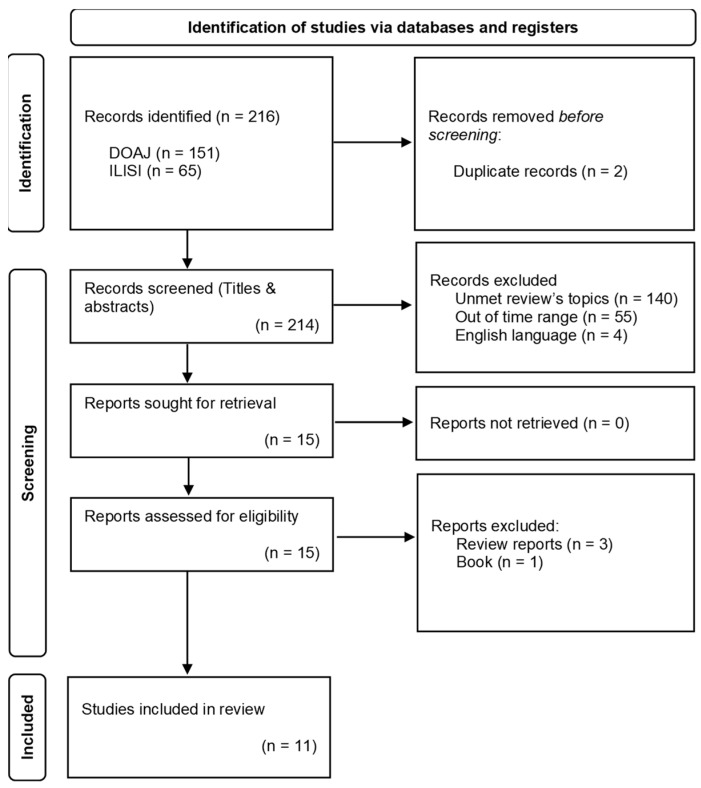
PRISMA flow diagram adapted from [34]. For more information, visit: http://www.prisma-statement.org/ (accessed on 15 March 2023).

**Table 1 healthcare-11-02573-t001:** Search strings and number of records found.

Database	Search Strings	Number of Results
DOAJ	[infermier* AND (assistenz* OR soddisfazione OR lavoro)] [nurs* AND (“nursing care” OR satisfaction OR workload)]	151
ILISI	[(“prestazioni assistenziali” OR “soddisfazione lavorativa” OR “carico lavorativo”) AND infermier*] [(“nursing care” OR “work satisfaction” OR workload) AND nurs*]	65

**Table 2 healthcare-11-02573-t002:** EPHPP quality rating. Adapted from [35]. For more information, visit: https://www.ephpp.ca/quality-assessment-tool-for-quantitative-studies/ (accessed on 15 March 2023).

Components	Quality Rating		
	Strong	Moderate	Weak
Selection bias	Very likely to be representative of the TP and ≥80% PR	Somewhat likely to be representative of the TP and 60–79% PR	All other responses or not stated
Study design	RCT and CCT	Cohort, case–control, interrupted time series	All other designs or not stated
Confounders	Controlled ≥80%	Controlled for 60–79%	Not controlled for or not stated
Blinding	Outcome assessor and study participants to intervention status and/or research question	outcome assessor or study participants	No blinding
Data collection methods	Tools are valid and reliable	Tools are valid but reliability is not described	No evidence of validity or reliability
Withdrawals and drop-outs	Follow up rate ≥ 80%	Follow-up rate of 60–79%	Follow-up rate of <60% not described

Legend: TP = Target Population; PR = Participation Rate; RCT = Randomized Controlled Trial; CCT = Controlled Clinical Trial.

**Table 3 healthcare-11-02573-t003:** Description of included papers.

First Author, Journal, Year. Title (IT/EN)	DB	Topic Aim of the Study	Type of Study, Sample Size, Tools	Results
D’Auria et al., NSC Nursing, 2017 [37] IT—Indagine osservazionale prospettica per la rilevazione del nursing workload in una unità di cure intensive pediatriche EN—Perspective observational survey to evaluate nursing workload in a pediatric intensive care unit	DOAJ	Nursing workload Nursing workload assessment in pediatric ICU	Prospective observational N = 51 Patients NAS	Workload indicators detention: - 8.1 days mean duration of ICU stay - 56.4% bed occupancy rate - 0.23 bed rotation index - 6.25 h mean working time per day NAS score (care complexity): 4% “Very High”, 25% “High”, 51% “Median”, 20% “Low”
Primavera & Leonelli, NSC Nursing, 2020 [38] IT—Un’indagine sulla percezione del carico assistenziale tra gli infermieri italiani, nell’era del COVID-19 EN—A survey on Italian nurses’ perception of workload in the COVID-19 era	DOAJ	Nursing workload To explore nurses’ workload perception during the COVID-19 pandemic.	Multicenter Cross-sectional N = 281 Nurses Questionnaire (ad hoc) IDA + (ICC)	281/956 (29.4%) completed questionnaires. IDA + (ICC) score: - 102 (36.3%) High dependency - 138 (49.1%) Median dependency - 41 (14.6) Low dependency
Miele & Pentella, NSC Nursing, 2016 [39] IT—La qualità dell’assistenza infermieristica in cure palliative: uno studio osservazionale EN—The quality of nursing care in palliative care: an observational study	DOAJ	Quality of care CGs’ satisfaction evaluation one month after patient’s death in palliative care setting	Cross sectional N = 150 CGs FAMCARE2	150/390 (38.5%) of completed questionnaires. FAMCARE2: - Patient’s dignity respect: 98% satisfied - Team availability: 96% satisfied - Information: 88% satisfied - Daily basic care: 88% satisfied
Arnone & Vicario, NSC Nursing, 2020 [40] IT—Benessere organizzativo e soddisfazione lavorativa: studio cross-sectional in una popolazione di infermieri EN—Organizational well-being and job satisfaction: cross-sectional study in the nursing population	DOAJ	Job satisfaction To assess perceived organizational well-being level and job satisfaction and to identify organizational factors influencing nurses’ well-being	Cross-sectional N = 318 Nurses Questionnaire (ad hoc), MBI	115/318 (36.2%) of completed questionnaires: - 76.5% unsatisfied with their job conditions - Work perception: - 42.6% interesting - 33.9% complex
Cosentino et al., Scenario, 2018 [41] IT—Le determinanti della soddisfazione lavorativa degli infermieri in area critica EN—Nurses’ job satisfaction-related factors in ICU setting	ILISI	Job satisfaction The aim of this study was to identify organizational factors that influence nurses’ job satisfaction in Emergency Department and High Care Department	Multicenter Cross-sectional N = 308 Nurses Questionnaire (ad hoc), MMSS, ELQ, CS, IWS, NPCS	308 completed questionnaires. 4 predictive factors (35% of overall JS): - nurse/physician collaboration (20%) - head nurse leadership style (10%) - compassion satisfaction (3.4%) - autonomy (0.9%).
Ziello et al., NSC Nursing, 2021 [42] IT—La soddisfazione lavorativa come indicatore di qualità per il dirigente delle professioni sanitarie: uno studio osservazionale EN—Job satisfaction as quality index for nursing managers: an observational study	DOAJ	Job satisfaction To assess the level of nurses’ job satisfaction in 2 hospitals in Rome	Multicenter cross-sectional N = 264 Nurses Questionnaire (ad hoc) and Stamps’s IWS	264/504 (52%) of completed questionnaires. 7 dimensions investigated [mean(±SD)]: - Autonomy = 42.4(±7.7) - Professional status = 33.3(±7.2) - Interaction with colleagues = 25.4(±5.6) - Organizational policy = 20.5(±5.7) - Task requirements = 19.6(±5.9), - Interactions with doctors = 18.9(±5.5) - Pay = 14.2(±6.5). Main job satisfaction-influencing factors: 3ys turnover (*p* = 0.002), work contract (*p* = 0.004).
De Simone et al., Mondo Sanitario, 2015 [43] IT—La qualità della vita lavorativa nelle organizzazioni sanitarie: risultati di un’indagine empirica EN—Quality of working life in Healthcare Organizations: Results of an empirical survey	ILISI	Job satisfaction Identification of factors influencing nurses’ job satisfaction and evaluation of its variations due to organization changes (new hospital building)	Cross-sectional (pre–post) N = not reported. Questionnaire (ad hoc)	63% of nurses in a not specified multiprofessional cohort: - no significant differences in healthcare workers’ satisfaction after organization changes - authors concluded that job satisfaction cannot be influenced by the organization changes where the directors’ leadership style was unchanged
Gnani et al., Italian Journal of Nursing, 2018 [44] IT—Indagine conoscitiva nazionale: la soddisfazione, il riconoscimento del ruolo infermieristico e il turnover degli infermieri giovani in medicina EN—National knowledge survey: satisfaction, recognition of the nursing role and turnover of young nurses	ILISI	Job satisfaction Job satisfaction recognition in young nurses, identification of factors influencing the decision to leave the medical ward	Multicenter cross-sectional (Mixed method) N = 144 Nurses	- 20% of the sample intended to leave the medical ward - 87% of the sample would suggest working in medical setting - 60% of the sample was satisfied (35% very satisfied) Factors influencing the decision to leave (qualitative data) were: professional recognition, teamwork quality, pay, workload, professional growing, and autonomy.
Incarbone et al., Professioni Infermieristiche, 2021 [45] IT—Impatto della job satisfaction sul turnover infermieristico: uno studio mixed-method. EN—Impact of job satisfaction on nursing turnover: a mixed method study	ILISI	Job satisfaction Job satisfaction recognition and exploration of factors influencing the decision to leave the ward	Mixed methods N = 133 Nurses (8 interviews) Questionnaire/interview	- 61% of the sample was satisfied, 21% unsatisfied - Highest degree of satisfaction was reported by nurses working in emergency settings while lower satisfaction was reported by surgical ward nurses. - Nurses over 50 years were more resistant to changing.
Marino & Vitale, Mondo Sanitario, 2017 [46] IT—Valutazione del grado di soddisfazione lavorativa degli infermieri italiani: una prospettiva di cambiamento EN—Evaluation of Italian nurses’ job satisfaction degree: A perspective of change	ILISI	Job satisfaction Job satisfaction assessment and identification of interventions to enhance it	Multicenter Cross-sectional N = 120 Nurses Questionnaire (ad hoc)	Participant-reported issues: - high degree of dissatisfaction - poor career opportunities - 79.1% reported too much responsibility - 78.6% lack of autonomy - 75% would became nurses again
Talucci et al., Professioni Infermieristiche, 2015 [47] IT—Empowerment strutturale e soddisfazione sul lavoro tra gli infermieri coordinatori: uno studio pilota EN—Structural empowerment and job satisfaction among head nurses: a pilot study	ILISI	Job satisfaction Head nurse job satisfaction assessment	Multicenter cross-sectional N = 125 Head Nurses Questionnaire (ad hoc)	Head nurses-reported issues: - Job dissatisfaction - Moderate structural empowerment The structural empowerment significantly influenced head nurses’ job satisfaction (*p* < 0.001).

Legend: DB = Database; IT = Italian; EN = English; DOAJ = Directory of Open Access Journal; ICU = Intensive Care Unit; N = Number; NAS = Nursing Activities Score; IDA = Indice di Dipendenza Assistenziale (Dependency to care index); ICC = Index of Caring Complexity; CGs = Caregivers; FAMCARE-2 = Family Satisfaction with Advanced Cancer Care-2; MBI = Maslach Burnout Inventory; ILISI = Indice della Letteratura Italiana di Scienze Infermieristiche (Italian literature on nursing science index); MMSS = McCloskey Mueller Satisfaction Scale; ELQ = Empowering Leadership Questionnaire; CS = Compassion Satisfaction scale; IWS = Index of Work Satisfaction; Nurse Physician Collaboration Scale.

**Table 4 healthcare-11-02573-t004:** Quality assessment—EPHPP score [35]. For more information, visit: https://www.ephpp.ca/quality-assessment-tool-for-quantitative-studies/ (accessed on 15 March 2023).

Author, Year	EPHPP Scores												
	SB		D		C		B		DC		DO		Overall
	R1	R2	R1	R2	R1	R2	R1	R2	R1	R2	R1	R2	
D’Auria et al., 2017 [37]	W	W	M	W	W	W	W	W	M	M	W	W	W
Primavera & Leonelli, 2020 [38]	W	W	W	M	W	W	W	W	W	M	W	W	W
Miele & Pentella, 2016 [39]	W	W	W	W	W	W	W	W	W	W	W	W	W
Arnone & Vicario, 2020 [40]	W	W	W	W	W	W	M	M	M	W	W	W	W
Cosentino et al., 2018 [41]	M	W	W	W	M	M	M	W	M	W	M	M	W
Ziello et al., 2021 [42]	W	W	W	W	W	W	M	M	W	W	W	W	W
De Simone et al., 2015 [43]	W	W	W	W	W	W	M	W	W	W	W	W	W
Gnani et al., 2018 [44]	W	W	W	W	W	W	W	W	W	W	W	W	W
Incarbone et al., 2021 [45]	W	W	W	W	M	W	M	W	W	W	W	W	W
Marino & Vitale, 2017 [46]	W	W	W	W	W	W	W	W	W	W	W	W	W
Talucci et al., 2015 [47]	W	W	W	W	W	W	M	M	W	W	W	W	W

Legend: EPHPP = Effective Public Health Practice Project; SB = Selection Bias; D = Design; C = Confounding; B = Blinding; DC = Data Collection; DO = Drop-Out; R1 = Reviewer 1; R2 = Reviewer 2. Quality Assessment: W = Weak; M = Moderate.

## Data Availability

Data sharing not applicable. No new data were created or analyzed in this study.

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
