# Peer review of "Workload, Job Satisfaction and Quality of Nursing Care in Italy: A Systematic Review of Native Language Articles"

_healthcare, 2023, doi:10.3390/healthcare11182573_

Round 1
Reviewer 1 Report
Overall, this manuscript has the value of research and discussion. Some suggestions are as follows:
1. Nursing culture and work connotation, nurse-to-patient ratio, salary structure, etc. are different in different countries. When summarizing the literature, there should be an overall explanation for this part, so that readers can analyze the context more clearly.
2. The level of the hospital (such as medical center or teaching hospital, etc.) will also affect the research results. If possible, it is recommended to include the result analysis statement.
3. Discussions and suggestions should be deepened and broadened.
Reviewer 2 Report
General issues
The aim of the article is to investigate some content aspects between native and international literature, focusing on the quality of nurses’ performance [lines: 74-75]. Three primary factors affecting the quality of nurse performance have been identified, i.e. workload, quality of perceived nursing care, and job satisfaction [lines: 77-78]. It was also indicated that the aims of this work were to review available papers on Nursing Workload (NW), Job Satisfaction (JS) and Quality of Care (QoC) published in authors own native language (Italian), and to discuss the findings in the light of international literature [lines 106-109]. The authors have defined the stages of the research process [lines 113-115]. Two databases were used to determine the sample - the Directory of Open Access Journals and the Indice della Letteratura Italiana di Scienze Infermieristiche [lines: 115-117]. The focus is solely on publications in Italian covering research from January 2015 to November 2022 [lines: 130-132]. The PRISMA method was used to select articles for the research sample [lines: 134-135]. In total, 11 scientific papers were analyzed [lines: 152, 161-163].
Detailed issues
Line 76: Does it make sense to extract a subsection when there is only one.
Line 126: Incorrect subsection number - should be 2.1. and not 3.1.
Line 126: Does it make sense to extract a subsection when there is only one.
Line 134: No explanation of the acronym PRISMA, it was only explained in the synopsis.
Line 149: In the table, some of the search words are shown in quotation marks and some are not, this should be standardized.
Line 265: Why in articles [33, 38, 45, 46, 47, 49, 50] some information is grayed out.
Line 265: If results from dashes are listed, a colon should be used first.
Lines 149, 152, 265, 273: No indication of sources under tables and figures.
Line 457, 483, 496, 503, 568, 583, 590, 596, 602: No period at the end of the line.
Lines 571-576: Standardize line spacing between lines.
Summary
The research sample is very small, with only 11 publications in Italian [lines: 130-132, 152, 161-163]. Three main factors of nurse performance quality have been analysed, however only one factor has been analyzed in more publications - JS [lines: 164-166]. A very modest comparison of the results of Italian-language authors with the results presented in the foreign-language literature was made. In particular, there is a modest discussion of the results regarding the NW and JS factors. The layout of the work also needs to be improved, some sub-chapters are mislabeled and there is no conclusion with a summary. A quality study was conducted using the EPHPP score, but the results were not discussed in detail.
Reviewer 3 Report
The central theme of the work, with respect to analyzing the quality of Workload, job satisfaction and quality of care in nursing area, is important for the improvement of both nursing work and its reflection in patient care.
However, these are subjective aspects whose interest and richness derives precisely from asking the opinion of the protagonists themselves, in this case, the nursing staff. For this reason, although a systematic review can always provide a compilation of works on the subject, it would be much more desirable, to broaden the theoretical-scientific body, a more focused approach, as we say in its protagonists.
On the other hand, reducing the information analyzed to a specific language, at a time when globalization is one of the riches for scientific progress, is not considered the most appropriate, beyond the interest at the national level to know the quantity, not the quality, of studies published on the subject.
For this reason, the publication of this work in the format presented is not recommended, suggesting that the sample analyzed be expanded to include all published works, at least in the languages ​​with the greatest current scientific scope, such as English.
Reviewer 4 Report
Dear authors
I would like to thank you for giving me the opportunity to review the manuscript entitled “Workload, job satisfaction and quality of care in nursing area: A systematic review on articles in Italian language”. This study reviewed papers published in the Italian language about workload, job satisfaction, and quality of care. I have some suggestions to improve the presentation of your work. My comments are as follows:
- You can use recent following article regarding nursing workload in introduction and discussion section and provide discussion about mental and physical workload:
Nursing physical workload and mental workload in intensive care units: Are they related? (https://onlinelibrary.wiley.com/doi/full/10.1002/nop2.785).
- Nurses’ quality of work life can play important role in job satisfaction. I suggest this issue addressed in the introduction.
- Methods section should be cover following issues in a systematic reviews: Protocol and Registration, Search Process and Eligibility Criteria, Study Selection process, Quality Appraisal, Data Collection Process and Synthesis of Results
- The limitation of the review should be added.
- Your study what messages have for nursing managers and nurse policy makings?
- Conclusion should be added.
Minor editing of English language required
Round 2
Reviewer 2 Report
General issues:
The authors introduced numerous changes, taking into account the comments and suggestions indicated in the review. The authors identified a new research question which was defined as: “How many Italian articles are present on job satisfaction, workload and quality of nursing? [lines: 156-157]. The method of identifying the research question has been changed from the PICO method to the PIOS method [lines: 161-162]. The chapter on research methods has been expanded, additionally separating important sections, i.e. Inclusion criteria and eligibility, Search strategy etc. [lines: 150-218]. An important element is the inclusion of an explanation of the adopted quality levels, which have been indicated in Table S1 [lines: 201-203]. The authors have extended the chapter to include a discussion [lines: 340-521]. Added chapter on Strength and limits [lines: 510-521]. Conclusions section added [lines: 522-529].
Detailed issues:
Lines: 171, 184: Shouldn't the indicated paragraphs start with a paragraph?
Lines: 194-195: The space between the subsection heading and the paragraph is smaller than otherwise.
Lines: 200-201: 6 or 7 fields are included. The text mentions 7 elements: selection bias, study design, confounders, blinding, data collection methods, withdrawals and drop-outs. Unless "withdrawals and drop-outs" is treated as one category, but then it's worth making it clear.
Line: 216: Should be 'Figure', not 'igure'.
Line: 330: In Table 1 under Incarbone et al. Part of the text marked in grey.
Line: 337: Why did item [45] get an overall score of 'W' when it scored 'M' in most categories?
Lines: 413-509: The font is smaller than the rest of the text.
Summary:
At the outset, it should be noted that the structure of the work has been improved. In particular, it is important to improve the division of work into chapters and subchapters. It is very good that the authors have expanded the chapter on the methodology of the study. In particular, it was important to discuss the adopted materiality levels. It is worth considering whether it would not be better to include Table S1 in the main content, and not in an external attachment. The extension of the discussion chapter, in particular the addition of 'Strength and limits', should be assessed positively. It is also very important that a chapter on conclusions has been added. It should be noted, however, that it is quite poor in content and conclusions. It would be worth expanding it. As in the first review, it should be pointed out that the research sample is very small, in fact only 11 publications. Additionally, publications were limited to Italian-language articles only. The authors try to justify such selection of the sample, however, it does not seem very convincing. Perhaps it would be worth extending the sample to English-language articles about Italian nurses. It also seems problematic to analyze factors that are marginally indicated in the Italian-language literature (1 or 2 publications). The article has been improved to a large extent, but still needs to be refined.
Author Response
Manuscript: Healthcare-2494125
Response to Reviewer
General issues:
The authors introduced numerous changes, taking into account the comments and suggestions indicated in the review. The authors identified a new research question which was defined as: “How many Italian articles are present on job satisfaction, workload and quality of nursing? [lines: 156-157]. The method of identifying the research question has been changed from the PICO method to the PIOS method [lines: 161-162]. The chapter on research methods has been expanded, additionally separating important sections, i.e. Inclusion criteria and eligibility, Search strategy etc. [lines: 150-218]. An important element is the inclusion of an explanation of the adopted quality levels, which have been indicated in Table S1 [lines: 201-203]. The authors have extended the chapter to include a discussion [lines: 340-521]. Added chapter on Strength and limits [lines: 510-521]. Conclusions section added [lines: 522-529].
Detailed issues:
[Reviewer]: Lines: 171, 184: Shouldn't the indicated paragraphs start with a paragraph?
[Reply]: Thank you for your suggestion. At line 170 and 183 we indicated the titles of the subparagraphs of the Methods paragraph: “2.2 Search strategy” and “2.3 Study selection and data extraction”
[Reviewer]: Lines: 194-195: The space between the subsection heading and the paragraph is smaller than otherwise.
[Reply]: Thank you for your suggestion. We changed
[Reviewer]: Lines: 200-201: 6 or 7 fields are included. The text mentions 7 elements: selection bias, study design, confounders, blinding, data collection methods, withdrawals, and drop-outs. Unless "withdrawals and drop-outs" is treated as one category, but then it's worth making it clear.
[Reply]: Thank you for your comment. To make the sentence more understandable we have used double quotes to indicate the six domains.
[Reviewer]: Line: 216: Should be 'Figure', not 'igure'.
[Reply]: Thank you for your suggestion. We corrected it
[Reviewer]: Line: 330: In Table 1 under Incarbone et al. Part of the text marked in grey.
[Reply]: Thank you for your suggestion. We unmarked it
[Reviewer]: Line: 337: Why did item [45] get an overall score of 'W' when it scored 'M' in most categories?
[Reply]: Thank you very much for your comment. The EPHPP provides instruction for grading the quality of multiple studies. At lines 205-208 we reported: “a standardized guide and dictionary are available for the rating system (Table S1). The assessment result was compared and discussed between the reviewers until a final decision on the global score was reached.”
In Addition, on Table 1 title (ex S1) the web link for the scoring system explanation it was reported as requested by the Reviewers. The above mentioned link https://www.ephpp.ca/quality-assessment-tool-for-quantitative-studies/ allows to readers to look the instruction.
In the specific case [item 45], considering that two or more Weak ratings on six dimensions of the EPHPP produces a Weak global score while one Weak rating is considered as Moderate, the 2 reviewers of the group were in contrast. The conflict was resolved discussing the issue with a third reviewer, as explained at lines 207-209.
[Reviewer]: Lines: 413-509: The font is smaller than the rest of the text.
[Reply]: Thank you for your suggestion. We changed the font to 10
[Reviewer]: At the outset, it should be noted that the structure of the work has been improved. In particular, it is important to improve the division of work into chapters and subchapters. It is very good that the authors have expanded the chapter on the methodology of the study. In particular, it was important to discuss the adopted materiality levels. It is worth considering whether it would not be better to include Table S1 in the main content, and not in an external attachment.
[Reply]: Thank you for your comment. Table S1 has been added in main text as Table 1. Other Tables were renumbered.
[Reviewer]: The extension of the discussion chapter, in particular the addition of 'Strength and limits', should be assessed positively. It is also very important that a chapter on conclusions has been added. It should be noted, however, that it is quite poor in content and conclusions. It would be worth expanding it. As in the first review, it should be pointed out that the research sample is very small, in fact only 11 publications. Additionally, publications were limited to Italian-language articles only. The authors try to justify such selection of the sample, however, it does not seem very convincing. Perhaps it would be worth extending the sample to English-language articles about Italian nurses. It also seems problematic to analyze factors that are marginally indicated in the Italian-language literature (1 or 2 publications). The article has been improved to a large extent, but still needs to be refined.
[Reply]: Thank you for your comment. Conclusion and Discussion section were improved in the manuscript as reported in previous version. About the publications limited to Italian-language only, we underline that the goal of this study was (line 145-149): “This work aimed to review available papers on Nursing Workload (NW), Job Satisfaction (JS) and Quality of Care (QoC) published in the authors’ native language (Italian), and to discuss the findings in the light of international literature. This could provide visibility to studies whose results on the quality of nurses’ performance may have greater dissemination and recognition.”
In other words, we have voluntarily focused our attention on articles published in Italian language only, because we wanted to have a complete overview of the topics we covered. In this way we can shows the effective status of quality of nurses’ performance in Italy. Many review papers (also including works published by Italian authors) were available in international databases while posed the use of English language as including criteria. About this, think on how much literature was reported using Chinese, Spanish, Portuguese, and Japanese languages today. We though that this can be a limit for review process, and it will be a growing issue.
About the sample size, finding works written only in Italian has been very difficult, as there are few databases and not a large national literature availability. However, many systematic reviews included few articles depending on selection criteria. For example:
-
Hafsteinsdóttir TB, van der Zwaag AM, Schuurmans MJ. Leadership mentoring in nursing research, career development and scholarly productivity: A systematic review. Int J Nurs Stud. 2017 Oct;75:21-34. doi: 10.1016/j.ijnurstu.2017.07.004. Epub 2017 Jul 6. PMID: 28710936. (15 studies)
-
Latina R, Iacorossi L, Fauci AJ, Biffi A, Castellini G, Coclite D, D’Angelo D, Gianola S, Mari V, Napoletano A, et al. Effectiveness of Pre-Hospital Tourniquet in Emergency Patients with Major Trauma and Uncontrolled Haemorrhage: A Systematic Review and Meta-Analysis. International Journal of Environmental Research and Public Health. 2021; 18(23):12861. https://doi.org/10.3390/ijerph182312861 (4 studies)
-
Biffi A, Porcu G, Castellini G, Napoletano A, Coclite D, D'Angelo D, Fauci AJ, Iacorossi L, Latina R, Salomone K, Iannone P, Gianola S, Chiara O; Italian National Institute of Health Guideline Working Group. Systemic hemostatic agents initiated in trauma patients in the pre-hospital setting: a systematic review. Eur J Trauma Emerg Surg. 2023 Jun;49(3):1259-1270. doi: 10.1007/s00068-022-02185-6. Epub 2022 Dec 16. PMID: 36526811; PMCID: PMC10229449. (5 Studies)
-
Castellini G, Gianola S, Biffi A, Porcu G, Fabbri A, Ruggieri MP, Coniglio C, Napoletano A, Coclite D, D'Angelo D, Fauci AJ, Iacorossi L, Latina R, Salomone K, Gupta S, Iannone P, Chiara O; Italian National Institute of Health guideline working group on Major Trauma. Resuscitative endovascular balloon occlusion of the aorta (REBOA) in patients with major trauma and uncontrolled haemorrhagic shock: a systematic review with meta-analysis. World J Emerg Surg. 2021 Aug 12;16(1):41. doi: 10.1186/s13017-021-00386-9. PMID: 34384452; PMCID: PMC8358549. (11 Studies)
-
Latina R, Salomone K, D'Angelo D, Coclite D, Castellini G, Gianola S, Fauci A, Napoletano A, Iacorossi L, Iannone P. Towards a New System for the Assessment of the Quality in Care Pathways: An Overview of Systematic Reviews. Int J Environ Res Public Health. 2020 Nov 20;17(22):8634. doi: 10.3390/ijerph17228634. PMID: 33233824; PMCID: PMC7699889. (9 Studies)
Reviewer 3 Report
The article, although it presents a subject of interest, uses a partial and unrepresentative image of the situation. There is no point in analyzing articles published in a specific language, outside the borders of said country, if it is not a comparative language or a language of special scientific interest because it is a minority or presents some interesting feature. Methodologically, the work is correct but could be improved to present greater interest as a systematic review that does not add novelty to the methodology already used with relish in the scientific field. For all these reasons, it is not considered a work of interest and relevance to the scientific community and its publication in these terms is not considered necessary.
Author Response
Manuscript: Healthcare-2494125
Response to the Reviewer
Comments and Suggestions for Authors
The article, although it presents a subject of interest, uses a partial and unrepresentative image of the situation. There is no point in analyzing articles published in a specific language, outside the borders of said country, if it is not a comparative language or a language of special scientific interest because it is a minority or presents some interesting feature. Methodologically, the work is correct but could be improved to present greater interest as a systematic review that does not add novelty to the methodology already used with relish in the scientific field. For all these reasons, it is not considered a work of interest and relevance to the scientific community and its publication in these terms is not considered necessary.
[Reply]: Thank you for your comment. The reviewer considers that the topics covered by articles published only in Italian language cannot contribute to the national framework of nursing research about Nursing Workload (NW), Job Satisfaction (JS) and Quality of Care (QoC). We understand the Reviewer’s concerns; however, the Italian situation could be of interest for national or international readers who consider the national differences as interesting issues. In addition, the authors believe that since nursing research in Italy is undergoing strong growth, nurses who undertake the path of scientific research face many difficulties. Many researchers due to lack of funds or scientific experience are forced to publish their results in local journals. However, their results could complete the overview already provided by some Italian studies published in English language, but which obviously do not include minor studies. In fact, these could help to clarify the Italian situation about nursing research. We hope the Reviewer will understand the effort made to bring at light studies that complement the information on the Italian framework of nursing research.
Reviewer 4 Report
Dear authors
Thank you for addressing my comments. I think the quality of your work much improved.
Author Response
Manuscript Healthcare-2494125
Comments and Suggestions for Authors
Dear authors
Thank you for addressing my comments. I think the quality of your work much improved.
[Reply]: Thank you very much for your suggestions.